# LISTENING TO FORMULAS: PIONEERING MODELS AND DATASETS FOR CONVERTING SPEECH TO LATEX EQUATIONS

## ABSTRACT

Recognizing spoken mathematical expressions is a challenging task that involves transcribing speech into a strictly structured symbolic representation while addressing the ambiguity inherent in the pronunciation of equations. Although significant progress has been achieved in both automatic speech recognition (ASR) and language models (LM), the specific problem of translating spoken formulas into LaTeX has received relatively little attention. This task is particularly important in educational and research domains, for example, for lecture transcription. To address this issue, in this paper, we present a pioneering study on Speech-to-LaTeX conversion, introducing a novel, diverse human-uttered dataset in English and Russian comprising 16000 (10000 in English and 6000 in Russian) distinct spoken equations uttered by three different speakers. Our approaches, which incorporate ASR post-correction and multi-modal language models, demonstrate a notable performance with up to a 25%

## 1 INTRODUCTION

Recent advancements in Automatic Speech Recognition (ASR) technologies, such as (Baevski et al., 2020b; Radford et al., 2023), have significantly improved the ability of these models to recognize spoken language. However, transforming spoken structured information, such as mathematical expressions, into symbolic formats like LaTeX remains largely unsolved. Current ASR systems, mostly pre-trained on a large plethora of unlabeled data in a self-supervised setting (SSL), can recognize some simple math symbols, such as $+$, $-$, or $\pi$, but cannot represent more complex equations. This limitation is especially critical given the growing demand for applications in academic, research, and educational settings, including the automatic transcription of mathematical content in lectures and the creation of accessible materials for individuals with hearing impairments. In this context, Speech-to-LaTeX (S2L) systems could serve as a powerful tool, enabling the transcription of spoken mathematical expressions into LaTeX for use in scientific documents, educational resources, and other structured content.

The primary challenge in the Speech-to-LaTeX task is that, unlike conventional ASR, it requires not only transcribing words but also understanding the hierarchical and nested structures inherent in mathematical notation. For example, the spoken phrase "the integral from zero to infinity" must be accurately transcribed into the LaTeX code `\int_0^{\infty}`, capturing both the verbal content and the underlying structure of the mathematical expression. This task involves more than just recognizing symbols; it requires an understanding of the relationships between components of mathematical statements, which are often non-linear and multi-dimensional in nature. Existing ASR systems, even those pre-trained on vast amounts of unlabeled data in a self-supervised setting, are not designed to handle this complexity due to the lack of specialized training data and models optimized for mathematical notation.

Transformer-based language models (LMs), such as BERT (Devlin, 2018), T5 (Raffel et al., 2020), GPT-3 (Radford et al., 2019), and more recently Qwen2-2.5B (Chu et al., 2023b), have demonstrated impressive capabilities, often surpassing human performance in natural language understanding tasks. Moreover, they can help to solve text-to-LaTeX tasks as shown in MathBridge (Jung et al., 2024) providing training data with textual pronunciation and LaTeX expressions as a target label.

Figure 1: (a) ASR post-correction and (b) Multimodal approaches generate a symbolic representation of LaTeX from spoken math expressions. In ASR post-correction, we feed audio into the model, extract the textual prediction, and then pass it to the LLM, which generates LaTeX. In the multimodal approach, we have two audio encoders, connect them to an adapter, add a prompt and feed it into Llama, obtaining the formula.

However, translating spoken mathematics into LaTeX code is still a relatively unexplored challenge. Addressing this gap requires not only new datasets but also the development of brand-new ASR systems fine-tuned to this unique task, potentially incorporating multimodal large language models and language models that can process both spoken input and handle mathematical content.

In this paper, we introduce the first comprehensive dataset designed specifically for Speech-to-LaTeX conversion in a bilingual setting. Our dataset comprises around 10k unique spoken equations in English and 6k in Russian, recorded by three different speakers among 40 data annotators (coverage rate is 3), ensuring variability in pronunciation, intonation, and linguistic style. To enhance the diversity of the dataset, we also include artificially generated audio using text-to-speech (TTS) models (Shen et al., 2018; Kong et al., 2020; Casanova et al., 2024) to increase the diversity of the data. This variety helps develop models that can generalize across different speaking patterns, making our dataset a good starting point for future S2L research.

To tackle the Speech-to-LaTeX challenge, we propose a hybrid approach that combines state-of-the-art ASR models (Radford et al., 2023; Chen et al., 2022a) with post-processing using fine-tuned language models and multimodal architectures (Tang et al., 2024; Sun et al., 2024; Chu et al., 2023a) capable of understanding both spoken language and text. While our Character Error Rate (CER) ranges from 6% to 45%, this variability is largely due to the inherent ambiguity in interpreting spoken mathematical expressions. For example, kappa" can be transcribed as either $\kappa$ or $\varkappa$ and the phrase "one over x plus 2" can correspond to several valid LaTeX representations such as $\frac{1}{x} + 2$, $\frac{1}{x+2}$, or $1/x + 2$. Despite these ambiguities, our system produces valid LaTeX expressions in most cases, establishing a strong baseline for future research.

Our contributions are threefold:

- **Dataset**. We introduce a high-quality open-source S2L dataset of spoken mathematical expressions in English and Russian, featuring diverse pronunciations and varying levels of complexity. This dataset provides a solid foundation for future research in multilingual Speech-to-LaTeX conversion. To the best of our knowledge, there are no existing datasets at the time of the writing.

- **Hybrid ASR and Audio-LLMs Approaches**. We introduce several architectures that combine ASR with LMs and multimodal LLMs to effectively translate spoken mathematics into LaTeX, addressing challenges of speech recognition and mathematical structure representation.

- **Evaluation and Benchmarking**. We conduct a comprehensive evaluation of our models using such metrics as CER, ROUGE-1, chrF and provide an in-depth analysis of the results, providing detailed analysis and establishing benchmarks for future work in this field.

## 2 RELATED WORK

**Automatic Speech Recognition Models**. Most ASR systems rely on spectrograms or mel-frequency spectrum input features instead of directly processing raw waveform to decrease the input

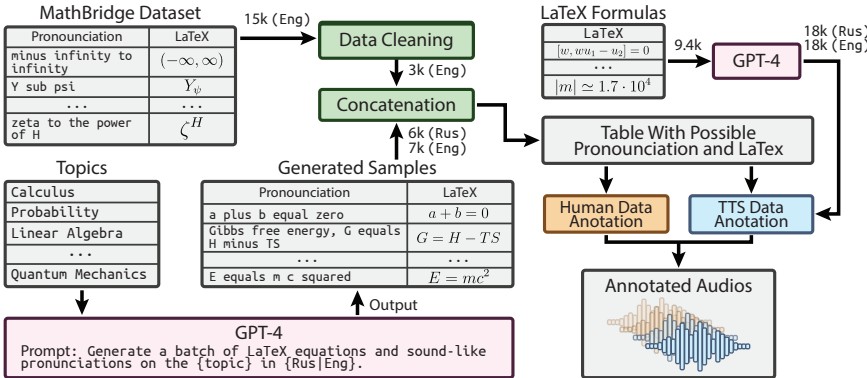

Figure 2: The dataset collection pipeline includes real data from MathBridge, LaTeX Formulas and synthetic data from GPT-4. For MathBridge, we've taken 15K samples, cleaned them, and got 3K samples. For GPT-4, we were asked to generate pronunciation-LaTeX pairs for English and Russian. All this data was labelled with TTS and Real Speakers. For LaTeX Formulas, we took 9.4K samples and asked GPT-4 to create four pronunciations for these formulas - two in English and two in Russian. This data was labelled with TTS only.

dimension. **Connectionist Temporal Classification (CTC)** (Graves et al., 2006; Amodei et al., 2016) loss function allows the models to align input speech sequences with output text without the need for the precise manual alignment of letters/phonemes with the corresponding audio (and consequently spectrogram) parts, forcing the model to learn the optimal alignment between audio frames and text sequences effectively. During inference, beam search is often used to maintain multiple leading hypotheses across different paths. However, CTC decoding operates independently of the previous context and independently from important semantic information. This problem might be mitigated by the attention mechanism (Luong et al., 2015; Vaswani et al., 2017). The **Listen, Attend and Spell (LAS)** model (Chan et al., 2016) adopts an encoder-decoder structure, where the encoder captures the input speech, and an attention mechanism allows the decoder to selectively focus on various segments of the input sequence as needed. The **Conformer** model (Gulati et al., 2020), on the other hand, combines convolutional neural networks (CNNs) with Transformer layers, thereby capturing both local features via convolution and long-range dependencies through self-attention mechanisms. **Wav2Vec 2.0** (Baevski et al., 2020a) employs a self-supervised learning approach to pre-train the model on unlabeled speech data using contrastive learning, learning high-quality representations from raw audio waveforms. **WavLM** extends the capabilities of Wav2Vec 2.0 by incorporating masked predictive learning and noisy student training, allowing it to handle speech recognition tasks in noisy environments more robustly. **Whisper** leverages a transformer-based architecture optimized in a weak-supervised regime and focuses on robustness and generalization ability to different languages and audio conditions.

**Language Models for Mathematical Understanding**. Many LLMs are specifically tuned for mathematical problems. For example, **Qwen2-Math** and **Qwen2.5-Math** (Yang et al., 2024) demonstrate remarkable performance in handling complex mathematical tasks in English and Chinese. It utilizes techniques like Chain-of-Thought (CoT) and Tool-Integrated Reasoning (TIR) to tackle complex problems. The model undergoes iterative self-improvement during training, leveraging synthetic data and reinforcement learning with a reward model. It is based on Qwen2-0.5B and Qwen2-2.5B, which have achieved state-of-the-art results on various natural language understanding tasks and have become a baseline for many language tasks due to the simplicity of fine-tuning. **ProofGPT-v0.1** is a 1.3B or 6.7B parameter language model based on the GPT-NeoX model and initialized with Pythia (Black et al., 2022; Biderman et al., 2023) weights. ProofGPT is tuned on the proof-pile dataset that consists of a collection of Arxiv papers. **Mathstral-7B-v0.1** LLM is a Mistral-7B model. On most mathematical reasoning benchmarks, it outperforms **DeepSeekMath-7B** (Shao et al., 2024), which uses supervised fine-tuning and direct preferences optimization (DPO). **Bumblebee-7B** is based on the Mistral model, tuned on the MetaMathQA dataset. (Yu et al., 2023). **InternLM-7B** (Ying et al., 2024) is also a commonly used model.

**Audio-LLMs**. Multimodal Language Models (MLLMs) have recently emerged. Their main idea is to transform the input modalities into embedding and properly combine them for further simultaneous usage in the LM subpart of the MLLMs for the next token prediction. Audio-LLMs such as SALMONNn and Qwen-Audio aim to bridge the gap between audio inputs and text-based language understanding. **SALMONNn** Tang et al. (2024); Sun et al. (2024) concatenates Whisper and BEATs (Chen et al., 2022b) (music perception model) embeddings, transform it with the Q-former (Li et al., 2023) and proceed to LLaMA-based LLM (Touvron et al., 2023) with the embeddings of the text instruction prompt. SALMONNn is trained to perform ASR, audio-based storytelling, and speech audio co-reasoning tasks. **Qwen-Audio** is a multi-task language model that extends Qwen's capabilities to audio-based inputs. The model was tuned to around 30 tasks, such as ASR, speaker recognition, and audio captioning, to achieve this quality. For the audio encoder, Qwen-Audio applies Whisper-Large. Following the multi-task training template proposed by Whisper and other multi-task models, it utilizes several special tokens (tags) to specify the task, audio and text languages, timestamps and transcription requirements.

**OCR Approaches for LaTeX Transcription**. In contrast to speech LaTeX recognition, image LaTeX (optical character recognition, OCR) recognition is widely studied in academia. Such open-source methods as Nougat (Blecher et al., 2023), pix2tex, im2latex, Textify, and TexTeller demonstrate good and robust results. OCR-LaTeX models can use techniques similar to the ASR models, such as CTC-loss and beam search decoding. OCR-LaTeX methods utilize encoder-decoder architectures with attention mechanisms to capture spatial and sequential dependencies in the input. For example, textify utilizes SWIN Visual Transformer (Liu et al., 2021) for the encoder.

**Post-Correction Techniques**. Post-correction (or post-processing) approaches are used to improve ASR transcriptions. Post-correction can employed to refine the output of ASR systems, particularly in text-to-LaTeX tasks, where authors fine-tuned T5, BART (Lewis et al., 2019; Liu et al., 2020), and GPT-3.5 in a supervised manner to transform the plain pronunciation-like text into the equation code on the proposed `MathBridge` corpus of LaTeX equations in context. Although this dataset contains millions of rows, the quality of the examples is low. For instance, equations and pronunciation are often repeated or do not match (the pronunciation describes something different). Moreover, there is a lack of long and difficult formulas, especially of good quality. Nonetheless, this work provides an important baseline for research on LaTeX processing topics.

**Datasets**. Textual datasets containing mathematical expressions, proofs, and formula transcriptions play a critical role in training LLMs to handle mathematical reasoning and symbolic manipulation tasks. The `Proof-Pile` dataset includes mathematical research papers, formal proof libraries, and textbooks. It has become a standard dataset for pre-training models to understand complex mathematical reasoning and symbolic representations. The `Open-Web-Math` dataset (Paster et al., 2023) contains 14.7B tokens of deduplicated mathematical content (including LaTeX formulas) filtered from `Common Crawl` dataset with attention. These are robust training corpora for training LLMs for base mathematical understanding and for handling benchmark mathematical tasks. We considered the open-source OCR-LaTeX dataset `OleehyO/latex-formulas`, which contains more than 500000 pairs of images and LaTeX formulas. `Im2LaTeX-100K` dataset contains around 100000 pairs of formulas from different areas. `IBEM` dataset consists of digital STEM document images with bounding boxes around formulas, providing a good dataset for LaTeX detection and capturing. It is used to train the TexTeller model. The most relevant dataset for the S2L tasks is the MathBridge dataset for the Text-to-LaTeX problem. This dataset provides textual pronunciation of mathematical expressions with corresponding LaTeX code and a short left and right context information serving. However, the absence of an audio component and the poor quality of samples limits its applicability for S2L tasks.

Unfortunately, all these datasets do not provide the spoken pronunciation of the formulas, committing the problem of converting spoken mathematics into LaTeX. That is why we started our research with the dataset collection.

# 3 DATASET COLLECTION

## 3.1 MOTIVATION AND APPROACH.

The creation of a high-quality dataset for the Speech-to-LaTeX (S2L) task presents a significant challenge due to the complexity and precision required for annotating spoken mathematical content. Manual annotation of such data is labour-intensive and requires a deep understanding of mathematical notation, making the process costly and time-consuming. To address this, we adopted a semi-synthetic approach, combining human-annotated and artificially generated data to create a robust and diverse dataset. We started by collecting pairs of LaTeX equations and a possible pronunciation of formulas. This pronunciation is essential for further voice-over: it is helpful for human annotators and represents a reference pronunciation, and speakers do not have to be profoundly aware of mathematical notation; it is mandatory for artificial annotations as an input to TTS or voice-conversion (VC) models. Several equations with different reference pronunciations were utilized to increase sample ambiguity.

## 3.2 DATA SOURCES AND PREPARATION

We employed a three-step approach to create the dataset, utilizing both real-world data and synthetic data generated by large language models (LLMs) and text-to-speech (TTS) systems.

**GPT-4 Generated Data**. Inspired by the recent advancements in multimodal models, we used GPT-4 to generate pairs of LaTeX equations and their corresponding pronunciations. For each study topic (e.g., Calculus, Mechanics), we prompted GPT-4 to provide 50–100 examples. The topics for the English and Russian parts were slightly different. After generation, we used rigorous data cleaning to remove empty, irrelevant, or duplicate samples. This step produced 7k unique pairs in English and 6k in Russian, covering a broad spectrum of mathematical topics and complexity levels. One can find several examples of the topics, possible equations and pronunciations in the Appendix in Table **??**.

**MathBridge Dataset Integration**. Additionally, we incorporated a subset of the `MathBridge` datasets. The primary advantages of this dataset are its considerable size, comprising over 23 million examples, and the inclusion of additional contextual information for the formulas. However, one significant drawback lies in the quality of the examples. We employed data cleaning to enhance the dataset for voicing and model training purposes. Our initial step involved selecting 15,000 examples from the original dataset, concentrating on the "spoken_English" and "equation" columns, while eliminating duplicate entries from the formula column. We then refined this subsample further by removing instances containing the following types of errors: (i) text instead of a formula; (ii) formulas that do not compile in LaTeX; (iii) entries marked as "None" in the pronunciation column (C: None); (iv) duplicated pronunciation including both text and numbers (e.g., forty-two: 42); (v) commands describing the formula in the pronunciation column; (vi) mismatched pronunciations that do not correspond with the formula (for example, the model may confuse the number of zeros in "0.005," describing it as "zero point zero five"); and (vii) nearly duplicated formulas, such as $\cos(\alpha)$, $\cos(\beta)$, ..., $\cos(\omega)$. As a result of our cleaning efforts, we retained 3,000 high-quality pairs for further inclusion in the `S2L` dataset.

**OCR-LaTeX Dataset Integration**. To further enhance the diversity of the dataset, we incorporated the `OleehyO/latex-formulas` dataset, which includes a wide range of complex and non-trivial equations. We extracted 9,400 unique formulas from this dataset and utilized GPT-4 to generate four distinct pronunciations for each formula: two in English and two in Russian.

## 3.3 AUDIO ANNOTATIONS AND DATASET COMPOSITION

**Human Data Annotation and TTS Audio Generation**. The next step was to voice over these pairs. To make human-annotated audio, we utilized the crowd-sourced platform similar to Amazon Mechanical Turk, where the equation and the possible pronunciation were displayed to the speaker. Annotators for Russian and English parts were different and did not intersect.

Also, open-sourced models (Kong et al., 2020; Casanova et al., 2024) and API-access proprietary models were applied to make artificially annotated audios.

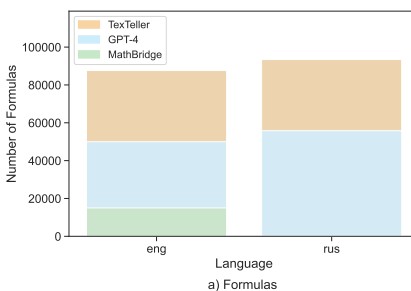 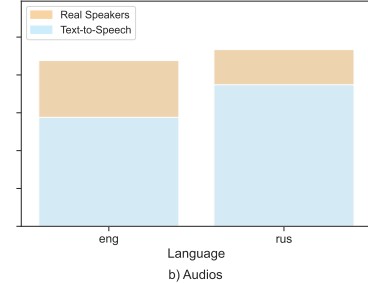

Figure 3: Distribution of language information in the dataset by data sources (a) and voices (b).

**Dataset Composition**. English part of S2L dataset consists of approximately 19.4K unique formulas, of which 3k are from the `MathBridge` set, 7k are generated using GPT-4, and 9.4k are obtained from `OleehyO/latex-formulas` (but number of unique pairs is 18.8k). 2 voices were used to annotate the English part: one was used as reference for the XTTSv2 and one from Russian TTS model. Overall, we have 57k+ English audios generated by TTS. We have a coverage of 3 speakers for humans, meaning three speakers voiced each formula from `MathBridge` and GPT-4, so that's about 30k audio recordings tagged with humans. The Russian part of the S2L dataset consists of approximately 6k examples generated using GPT-4 and 9.4K examples taken from `OleehyO/latex-formulas`, the same as for the English subset. We dubbed these 6k with 6 Russian TTS Rus voices and 18.8K examples from `OleehyO/latex-formulas` with similar to the Eng subset XTTSv2 and Russian TTS voices, resulting in 74k TTS-labeled audio recordings. Also, the GPT-4 equations were labelled by people with coverage of 3 speakers per formula, resulting in 18k human-annotated audios.

To ensure the high quality of the dataset, we conducted manual verification of the collected data. The general overview of the dataset creation process is presented in Figure 2. The distribution of data sources and voices is shown in Figure 3.

## 4 EXPERIMENTS

### 4.1 DATASET SPLITS

We considered several ways to make train-val-test splits for the evaluation. The primary way is to split our combined dataset into parts corresponding to non-overlapping parts of equations, meaning formulas from the test were not included in the train set, depriving the model of the opportunity to remember it. This was made to test LLMs and Audio-LLMs generalization abilities. The second way is to put all artificial audios into the train, and val sets while keeping human audios in the test set to check whether the artificial annotation, which can be considered as a pseudo-labelling technique, serves as well for generalization abilities to real-world data. In most experiments, we consider only human audio for the test set if not stated otherwise. The train set might combine human and artificial audio or only artificial ones. The validation set is distributed similarly to the train set. We combined TTS-generated audio recordings and human speech in the training and validation sets to create a more diverse dataset that improved the model's ability to generalize across different input data types, enhancing overall reliability and performance. We also considered monolingual and bilingual splits to verify whether cross-language training helps to perform better on the particular language test set subpart or whether training in a monolingual setting solely outperforms bilingual training. All pronunciations were striped.

### 4.2 EVALUATION METRICS

We consider several metrics commonly used in speech recognition, summarization and translation for the evaluation. The main metric is Character Error Rate (CER), which is defined as the ratio of the normalized edit distance (Levenshtein distance) between the predicted sequence and the ground truth: $\text{CER} = \frac{S+D+I}{N}$, where $S$ is the number of substitutions, $D$ is the number of deletions, $I$ is the number of insertions, and $N$ is the total number of characters in the reference. ROUGE-1

Table 1: Example of transcription

| Model | Transcription |
|---|---|
| Whisper Large-v3 | The covariant derivative of a vector a mu equals partial mu with respect to X nu plus gamma upper rho mu nu times a rho. |
| WavLM | the covarient derivative of a vector a mou equals partial moo with respect to ex new plus gama upper row moo new times a row |
| Wav2Vec2 | the covariant derivative of a vector a mu equals partial mo with respect to x-new plus scamma upper row mo new times a row |
| Qwen-audio | The covariant derivative of a vector mu equals partial mu with respect to x nu plus gamma upper row mu nu times a row |
| Canary | The covariant derivative of a vector amu equals partial amu with respect to x nu plus gamma upper rho moon nu times a rho. |

(Lin, 2004) calculates the unigram recall between the predicted output and the reference text. BLEU and sacreBLEU (Papineni et al., 2002; Post, 2018) evaluate n-gram precision by comparing the predicted output against the reference. chrF and chrF++ are character-based F-scores metrics that compute a balance between precision and recall at the character level. To more fairly evaluate the general understanding of the S2L models, predicted LaTeX formulas and ground-truth labels are transformed into lowercase before metric calculation if not stated differently. Its approach is valid, as capitalization is not indicated directly in most pronunciation and audio voice-overs.

### 4.3 ASR Post-Correction

The first approach to solving the S2L task is ASR post-correction (or post-processing). The ASR post-correction process is a method that combines two techniques in sequence: ASR and LLM. The first step is to use ASR to transcribe audio into text, and then the second step is to apply LLM to create a LaTeX formula representation of the transcript. Post correction is quite natural for this task, as it allows the LLM model, which has general mathematical knowledge, to transform the ASR output text into a specific structured format of the LaTeX. To achieve the same level of quality, a stand-alone ASR model should be trained on quite a large amount of audio data, which falls into the problem of supervised labelling of the audio data. Shallow (ASR + LLM hypothesis rescoring during inference) and deep fusion (simultaneous training of ASR and LLM) of ASR model with math-aware LM can help to achieve better results, but it has several drawbacks: inference decoding with large LM first pass rescoring would be highly time and memory consuming; deep fusion is hard to train, and it increases the complexity of the model. We attempted to train ASR-only Speech-to-LaTeX, but due to poor linguistic training, the model metrics were unsatisfactory, so this approach was abandoned.

We considered Qwen2, Qwen2.5, Qwen2.5-Math and ProofGPT for the LLMs options. This setup was trained and tested in English, Russian and English + Russian cases. Additionally, Flan-T5 Large (Chung et al., 2024) was tested on an English set only. In our experiments, we fine-tuned the entire model when the size was smaller than 7B. For the 7B model, we considered fine-tuning using LoRA with a rank of 4 and an alpha of 8. However, the experiments with LoRA were unsuccessful, as the model generated incoherent text for certain queries. We used Whisper Large-v3, Canary, and Wav2Vec 2.0 for Speech-to-Text transcription. Whisper and Canary provide the most appropriate transcription, while WavLM and Wav2Vec2.0 can make serious errors. Qwen-Audio also provides relatively good transcription (since it is based on Whisper Large-v2). See example of transcription $\nabla_\nu A^\mu = \frac{\partial A^\mu}{\partial x^\nu} + \Gamma^\mu_{\nu\rho} A^\rho$ `\nabla_\nu A^\mu = \frac{\partial A^\mu}{\partial x^\nu} + \Gamma^\mu_{\nu\rho} A^\rho` in Table 1.

## 4.4 MULTIMODAL MODELS

We applied the Qwen-Audio and SALMONN-13B models for Audio-LLM experiments due to their superior performance across various benchmarks. In this approach, audio encoders generate a hidden representation of the waveform, which is then passed to an adapter that converts it into a format compatible with LLM tokens. The resulting audio tokens are concatenated with system prompt tokens, and the combined sequence is fed into the LLM, which outputs the corresponding LaTeX formula. The LLM and adapter components of SALMONN are fine-tuned with different system prompts. The Qwen-Audio model was fine-tuned using LoRA, applied only to the LLM layers.

## 5 RESULTS AND DISCUSSION

We computed several metrics, described in Section 4.2, with the HuggingFace evaluate library. First, we introduce more character-centric metrics, such as CER and chrF.

Table 2 compares the performance of various language models on lower-case metrics across English, Russian, and combined English-Russian datasets. The table provides Character Error Rate (CER), Rouge-1, sBLEU, and chrF metrics. Among the models evaluated, SALMONN consistently achieves the best overall performance. In the English dataset, SALMONN leads with the highest Rouge-1 (83.88), sBLEU (60.68), and chrF (71.04) scores, though its CER (42.42) is slightly higher than Qwen2.5-Math-1.5B, which has the lowest CER (39.54) and ranks second in Rouge-1 (81.43) and chrF (68.34). For the Russian dataset, SALMONN again outperforms, with the best scores across all metrics, including CER (10.45), while Qwen2.5-Math-1.5B closely follows. In the combined English-Russian dataset, Qwen2.5-0.5B excels with the lowest CER (22.70) and highest Rouge-1 (86.22), sBLEU (67.14), and chrF (79.87), outperforming ProofGPT. Overall, SALMONN dominates in English and Russian, while Qwen2.5-0.5B shines in the combined dataset.

Table 3 presents the performance metrics for non-overlapping formulas across the training, validation, and test sets, comparing two versions of the Qwen models (Qwen2-0.5B and Qwen2.5-0.5B) for Russian and English languages, as well as a combined English and Russian dataset. The table reports the Character Error Rate (CER), Rouge-1, sBLEU, and chrF metrics, where CER indicates error rates (lower is better), and the remaining metrics reflect accuracy (higher is better). For both Russian and English languages, Qwen2.5-0.5B consistently outperforms Qwen2-0.5B in terms of Rouge-1, sBLEU, and chrF, particularly on the test set. Interestingly, in the case of the combined English and Russian datasets, the two models exhibit very close performance, with Qwen2.5-0.5B showing marginal improvements in accuracy metrics while having a slightly higher CER. Notably, the test set was voiced using real human speakers, contrasting with the text-to-speech (TTS) voicing applied to the training and validation sets, as highlighted in the table notes.

We also evaluated the success rate of compiling formulas into LaTeX - whether the formula compiles into LaTeX without errors or not. The models reached up to 95-99% compilation success rate.

Speech-to-LaTeX models can quickly convert spoken language into mathematical formulas. Unlike a human who needs to listen, interpret, and manually enter data, these models automate the entire process, significantly reducing the time it takes to complete a task. It is beneficial in environments where agility is essential, such as during lectures, conferences, or webinars. It also simplifies the process for those who dictate formulas, as they no longer have to wait for someone to transcribe them manually.

Additional metrics for lower-case performance can be found in Appendix in Table **??**, and for case-sensitive in Tables **??** and **??**, respectively.

We present the results of the SALMONN-13B generation, which show sufficient quality. There are also some limitations, which will be mentioned later in the paper. The metrics are generally relatively good, but sometimes, they do not reflect the actual situation. To assess the quality of generation, see Table 4

## 5.1 CROSS-LANGUAGE LEARNING

One of the advantages of fine-tuning multilingual language models is the ability to extract information from one language that is not available in another. For example, LaTeX special symbols

Table 2: Lower-case metrics for different Language Models

| Model | Language | CER ↓ | Rouge-1 ↑ | sBLEU ↑ | chrF ↑ |
|---|---|---|---|---|---|
| Qwen2.5-0.5B | Eng | 43.87 | 77.78 | 53.33 | 64.48 |
| Qwen2.5-Math-1.5B | Eng | **39.54** | 81.43 | 57.86 | 68.34 |
| ProofGPT-1.3B | Eng | 41.60 | 78.04 | 52.31 | 64.30 |
| SALMONN-13B | Eng | 42.42 | **83.88** | **60.68** | **71.04** |
| InternLM2-1.8B | Eng | 49.23 | 78.12 | 61.00 | 64.24 |
| Flan-T5 | Eng | 64.92 | 53.47 | 11.98 | 28.78 |
| Qwen-Audio | Eng | 52.66 | 76.63 | 57.78 | 60.96 |
| Qwen2.5-0.5B | Rus | 13.19 | 89.71 | 72.78 | 86.09 |
| Qwen2.5-Math-1.5B | Rus | 10.49 | 90.66 | 74.25 | 88.11 |
| ProofGPT-1.3B | Rus | 16.48 | 87.82 | 70.82 | 84.04 |
| SALMONN-13B | Rus | **10.45** | **93.59** | **76.63** | **91.63** |
| Qwen2.5-0.5B | Eng+Rus | **22.70** | 86.22 | 67.14 | 79.87 |
| ProofGPT-1.3B | Eng+Rus | 23.93 | 84.85 | 65.33 | 78.18 |
| SALMONN-13B | Eng+Rus | 24.27 | **89.93** | **69.62** | **84.10** |

Table 3: Metrics (%) results on non-overlapping formulas on train, validation and test sets.

| Model | Language | Test | CER ↓ | Rouge-1 ↑ | sBLEU ↑ | chrF ↑ |
|---|---|---|---|---|---|---|
| Qwen2-0.5B | Rus | Human | **7.09** | 94.44 | 79.59 | **92.79** |
| Qwen2.5-0.5B | Rus | Human | 7.49 | **94.58** | **79.88** | 92.73 |
| Qwen2-0.5B | Eng | Human | 25.05 | 86.56 | 70.39 | 76.91 |
| Qwen2.5-0.5B | Eng | Human | **23.56** | **86.92** | **71.37** | **77.88** |
| Qwen2-0.5B | Eng+Rus | Human | **30.36** | 83.52 | 61.72 | 72.20 |
| Qwen2.5-0.5B | Eng+Rus | Human | 31.13 | **83.60** | **61.73** | **72.22** |

`\simeq` and `\hat` are not presented in the Russian part of the dataset but in English. Qwen2.5, trained in English and Russian, can transcribe "approximately equal" in Russian to `\simeq` ($\simeq$). Another observation is that the models are mostly English-oriented, so Qwen2.5-Math-1.5B and Qwen2-0.5B trained in Russian can generate only simple formulas in English. The reverse situation works worse - Qwen2.5-0.5B, trained in English, cannot perform post-correction in Russian.

The second advantage is the performance. We fine-tune the model with multilanguage data and show whether this improves performance. To do so, we will use the benchmark Qwen2-0.5B trained in English+Russian and the results in English to see if they got better. See Table 5.

Analyzing the performance difference of the Qwen2-0.5B model trained on English data versus the combination of English and Russian data evaluated on the English test set, we can say that the model trained on both languages achieves better results in Rouge 1 (87.77 *vs.* 86.56), sBLEU (72.44 *vs.* 70.39), and chrF (79.01 *vs.* 76.91), indicating improved accuracy in capturing the structure and content of formulas. However, the CER increases slightly (26.27 *vs.* 25.05), suggesting a minor trade-off in transcription accuracy. It indicates that multilingual training can enhance the model's ability to generalize and improve formula representation, though it may slightly affect error rates. Another result of cross-language learning is presented in Table **??**.

## 5.2 LIMITATIONS

There are many exs where both predicted and Ground Truth LaTeX give the same formula, but a different code is used, leading to the metrics' degradation. For instance, when true LaTeX is `\int_{a}^{b} f(x) dx` and the model generates `\int_a^b f(x), dx`. Also, capital and non-capital letters are a challenge. LaTeX formula renders different letters and special symbols

Table 4: Examples of Generations for SALMONN.

| LaTeX | GT | Pronunciation |
|---|---|---|
| $F_{\mu\nu} = \partial_\mu A_\nu - \partial_\nu A_\mu$ | $F_{\mu\nu} = \partial_\mu A_\nu - \partial_\nu A_\mu$ | the field strength tensor for electromagnetism is F mu nu equals d mu A nu minus d nu A mu |
| $\int x\,dx = \frac{1}{2}x^2 + C'$ | $\int x\,dx = \frac{1}{2}x^2 + C'$ | the integral of x dx equals one half x squared plus C |
| $n(\mu, \sigma^2, t)$ | $\mathcal{N}(\mu, \frac{\sigma^2}{T})$ | N of mu, sigma squared over T. |

Table 5: Remaining metrics (%) results on non-overlapping formulas on train, validation and test sets.

| Model | Train Language | Test Language | CER ↓ | Rouge-1 ↑ | sBLEU ↑ | chrF ↑ |
|---|---|---|---|---|---|---|
| Qwen2-0.5B | Eng | Eng | **25.05** | 86.56 | 70.39 | 76.91 |
| Qwen2-0.5B | Eng+Rus | Eng | 26.27 | **87.77** | **72.44** | **79.01** |

depending on the case, like $\phi$ and $\Phi$. An additional pivot point in risks in metrics calculation is the symbol styles: `\mathcal{R}` and `r` can be pronounced similarly and mean the same, but CER between these codes is much larger than one.

As we already discussed, there are ambiguous examples, such as "2 squared from x plus 1" can be either $\frac{2}{x^2+1}$ or $\frac{2}{x^2} + 1$. One way to solve this problem is to say "parentheses" when necessary. In this case, all parts of the formula that need to be raised in degree or perform another operation will be separated by open and closed parentheses. Some `MathBridge` samples follow this strategy, but in most cases, the parentheses are ignored.

Another limitation is the ASR system. Our method primarily depends on the quality of the transcript. If the model produces an incorrect representation of a sound due to poor sound quality, specific pronunciation, or some background noise, we will not be able to generate a good formula. We can tune the ASR models to be more robust and train the LLM to recognize and correct these types of errors. These limitations will be considered in future research.

## 6   CONCLUSION

In this paper, we were introduces Speech-to-LaTeX, a novel speech conversion task. For this purpose, we collected 53k pairs of LaTeX equations with a possible pronunciation in English or Russian. Pronunciations were a reference for the human annotators and an input to the TTS models. The pairs were collected from 3 sources: (1) 3k from the MathBridge dataset (Eng), (2) 13k pairs (6k Rus and 7k Eng) were generated and pronunciated using open-source LLM on various physical and mathematical topics, and (3) 9.4K unique formulas were taken from the OCR-LaTeX dataset and pronounced four times (2 Eng and 2 Rus) automatically and differently, resulting in 37.6K pairs. Every pair from (1) and (2) was annotated by three random speakers among 33 annotators. Every pair from (1)-(3) was annotated with TTS at least twice. Our S2L dataset consists of 180K unique triplets of pronunciation-LaTeX-audio. We trained and evaluated different Audio-LLMs and ASR-LLM post-correction models. The SALMONNn and Qwen2.5-Math demonstrated the best performance regarding CER and ROUGE-1 metrics. The experiments showed good performance for Speech-to-LaTeX conversion and a benefit of cross-language learning. Overall, we expect this work to contribute to developing speech recognition research in the natural science domain and become a baseline for the Speech-to-LaTeX problem. Future work might be devoted to the additional dataset collection, especially annotation of lecture recordings, audio-visual S2L, and experiments combining text and equations.

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
