# OpenReview forum: "Listening to Formulas: Pioneering Models and Datasets for Converting Speech to LaTeX Equations"
_ICLR.cc/2025/Conference — ICLR 2025 Conference Withdrawn Submission_

### Official Review · Reviewer_hX9V · 2024-10-19

**Soundness:** 2
**Presentation:** 2
**Contribution:** 2
**Rating:** 3
**Confidence:** 4

**Summary:**

In this work, the authors focus on the speech to latex (S2L) task, which is not well explored. The first challenge is little relevant data available and authors created datasets in both English and Russian to address this issue. They also explore different modeling methods to tackle with the S2T based on the proposed datasets.  However, there are many information missing or not addressed well.

**Strengths:**

- A dedicated dataset is created for the S2L task. The dataset is from three sources: 1) GPT-4 generated latex samples based on prompt 2) MathBridge subset after filtering 3) OleehyO/latex-formulas dataset and generated with GPT-4

- Propose several methods to tackle with the S2L task

**Weaknesses:**

1.  It is great to create dataset for the new tasks. My main concern for the dataset is the coverage or representativeness of the dataset.
- Part of the data is generated by GPT-4 with prompt. The data is highly dependent on the model and prompts used.  There is no analysis conducted how representative the data is
- Part of the data is based on MathBridge, which is created based on arXiv papers. However, only 15,000 samples are selected from 23 million samples and 3000 of them are used after filtering. The selection rules and filtering methods could introduce bias too. There is no analysis conducted how representative the data is for the arXiv papers.
2. The paper lacks information for model fine-tuning, which is critical to compare different methods.

**Questions:**

- For the evaluation, do we evaluate the results based on the spoken form as displayed in Table 1? Why not use the written form (latex equation)?
- Do we use the data to fine-tune the acoustic models, such as Whisper, Canary, WavLM and wav2vec2? How about Qwen-Audio, which is built for speech and why it performs worse than other LLM only model?

---

> ### Author Response · Authors · 2024-11-22
>
> Thank you for the detailed review. Let us address weak points and answer your questions.
>
> **W1.**
>
> It is hard to evaluate how representative our data is. There are too many fields of mathematics and natural sciences and we do not pretend to cover them all. However, firstly, we were fostering to cover the main LaTeX mathematical symbols, commands and letters, such as \alpha … \omega, \frac{}{}, \sqrt{}, \left(, …, etc.  Secondly, we prompted the GPT to provide examples and pronunciation on topics that are typical for the BSc in math or physics, manually removing too simple samples or the samples that contain mainly the \text{} instead of special characters.   Thirdly, we used 3 different formula sources. Fourthly, we tried to voice-over the equations with several voices in two languages. Finally, part of the equation is pronounced in two ways: sounds-like and  name-like, e.g. (“f equals m a” and “Newton’s second law”). From these points of view, we assume our dataset to be representative. Moreover, we demonstrate that the Speech2Latex quality does not degrade significantly if we train the model on synthetically generated audios.  Consequently, we encourage anyone to generate examples that are relevant to his/her topic almost for free and tune the model. However, we should admit, that usually there is an ambiguity of pronunciations and we cannot cover them all in audio, leaving it to the language model to handle it (“f equals m a”, “f is equal m a”, “f equals m multiplied by a”, “Newton’s second law”, “Newton’s second law of the motion”, … -> $F=ma$ or $\vec{F}=m\vec{a}$).
>
> MathBridge was presented and became open-sourced during an intermediate stage of our work. However, if one looks closely at it, one can notice that it contains too many poor equations, as discussed in our work. We were physically unable to proceed with many formulas and considered 15,000 random samples. We removed the primary useless (too short, too many punctuation signs, too similar ones, etc.) and decided to switch to better-quality data.
>
> The most representative dataset is OleehyO/latex-formulas of OCR-formulas, as it contains many comprehensive examples from real papers. We better should have started with this dataset, but we did not realise to use it during the early stage of our work.
>
>
> **W2.**
>
> We will add this information to the manuscript, which will be presented in the updated version. So far, the information can be found in the supplementary materials and the code.
>
>
> **Q1.**
>
> We evaluate the results in the LaTeX form, indeed (see, for example, table 4, where “GT” represents the ground truth equations, and “LaTeX” represents the S2L model’s prediction). Table 1 illustrates how different ASR models transcribe one particular audio to support our claims of which models are more suitable for the S2L task.
>
> **Q2.**
>
> No, we did not fine-tune ASR models. In the first part of the experiments, we used pre-trained Whisper to transcribe audios and tuned LLMs in a post-correction setting (Whisper transcription to Latex). We fine-tuned them in the audio-to-latex setting for the second part of experiments with audio-LLMs (Qwen-Audio and SALMONN).
> The reason why Qwen-Audio performs worse than other methods is probably due to the nuances of re-implementations. Qwen-audio's official repository does not provide training or fine-tuning scripts, thus, we had to re-implement the fine-tuning of the model on our own, which could lead to a worse performance than claimed by the authors.
>
> We have made our dataset publicly (and anonymously) available [https://huggingface.co/datasets/marsianin500/Speech2Latex](https://huggingface.co/datasets/marsianin500/Speech2Latex).
>
>
> We hope you will find these explanations helpful. Please feel free to ask additional questions; we will be glad to answer them.

---

> > ### Comment · Reviewer_hX9V · 2024-11-25
> > **Thanks for the reply**
> >
> > W1
> > > It is hard to evaluate how representative our data is.
> >
> > I agree it is very hard but it is critical for a dataset to be used for others' work. One possible solution is to download latex files from arXiv and compare latex notations in your dataset with the real distribution.
> >
> > > too short, too many punctuation signs, too similar ones, etc.
> >
> > I am not sure "too similar ones" or " too short" could be an issue for the dataset. It might just reflects the data distributions in the real world.
> >
> > Q2
> > Based on the reply, I feel more work is required for the experimental section.

---

### Official Review · Reviewer_oZFx · 2024-11-04

**Soundness:** 3
**Presentation:** 4
**Contribution:** 2
**Rating:** 5
**Confidence:** 4

**Summary:**

This paper addresses the challenge of converting speech to LaTeX.

It introduces a 3-way dataset containing speech, text, and LaTeX formulas in two languages: English and Russian.

The study explores various model architectures, including hybrid ASR post-correction and multimodal audio-LLMs, achieving promising results across several metrics (CER, ROUGE-1, sBLEU, and chrF).

**Strengths:**

1. The paper is well-written and easy to read,  making it straightforward for readers to follow the authors' methods and findings.

2. A good contribution of the paper is the introduction of a new dataset for the speech-to-LaTeX task. This dataset, which includes speech, text, and LaTeX formulas in both English and Russian, is a valuable resource for the field. The authors also provide the introduction of the data collection process. This can be beneficial not only for replicating their work but also for other researchers looking to develop similar resources or expand on this dataset in future studies.

3. The paper includes a good introduction to related work, it provides readers with sufficient background knowledge.

4. The experimental section of the paper is thorough, comparing the performance of various models across multiple evaluation metrics, including CER, ROUGE-1, sBLEU, and chrF. By evaluating different model architectures, such as hybrid ASR post-correction and multimodal audio-LLMs, the authors demonstrate the robustness and effectiveness of their approach as well as making the findings more convincing and reliable.

**Weaknesses:**

Main weaknesses:
1. Lack of novelty: Apart from the newly introduced dataset, the paper’s contributions are fairly limited. Both hybrid ASR post-correction and audio-LLM adaptation are quite standard and are well-explored areas, and the paper does not present any particularly novel or innovative ideas from a model/pipeline perspective. Additionally, the results align with what might be expected.

2. The scope of this paper is somewhat narrow, as the speech-to-LaTeX task is a very specific subtask of Speech Translation. The practical relevance of this task is also unclear, as it’s difficult to find real-world scenarios where people speak only in formulas, even in math classes. A more practical approach might involve developing a model capable of distinguishing between natural speech and mathematical formulas within the same input. So that it can handle both seamlessly.

Minor weakness:

3. It would be better to include more analysis of the cases that failed to compile. In this paper, it was simply described as "The models reached up to 95-99% compilation success rate."

**Questions:**

1. Does Qwen-Audio support Russian?

2. There are missing table numbers in the following locations:
 - Line 242
 - Lines 422 and 423
 - Line 479

---

> ### Author Response · Authors · 2024-11-22
>
> Thank you for the detailed review. Let us address weak points and answer your questions.
>
> **W1.**
>
> Our main contributions are the dataset, the baseline training strategies, and the benchmarking for further research.
>
> Although different combinations of ASR and LMs were studied thoroughly, to our knowledge, there is a lack of work devoted to processing structured but ambiguous data as equation pronunciations. In contrast, many works are dedicated to the NLP methods for processing structured and ambiguous data, e.g.,  chemistry formulas processing or text2code/SQL.
>
> **W2.**
>
> As we indicate at the end of the conclusion, simultaneous processing of the equations and ordinary speech is more prominent in the application.
>
> The main problem is the absence of a suitable dataset for further voice-over. Mathbridge should have solved this issue as it also contains a short left and right context of the equation. Unfortunately, as discussed in the text, the Mathbridge dataset contains too many poor samples, making it hard to apply and annotate directly. We can use equations from a good-quality OleehyO/latex-formulas dataset, cover the equation with random artificial/GPT in the left and right contexts, and then annotate the obtained texts with TTS models. It should provide satisfactory results (as demonstrated in the text, trained on the TTS samples model performs relatively well compared to the model trained on the human data). Human annotation is more complex, expensive, and time-consuming, and we will probably use it for future work.  Another way is to parse the relatively long formulas with context from the Arxiv article latex codes. One of the new model's possible applications is paper writing assistance.
>
> **W3.**
>
> For example, issues with brackets. We will add an analysis of the failure cases to the revisited manuscript.
>
> **Q1.**
>
> No, Qwen-Audio does not support the Russian language.
>
> **Q2.**
>
> The mentioned uncertainties and miss-printings are fixed:
> - Line 214->242:  reference to the “Table 6”
> - Lines 416-420 moved to the introduction.
> - Lines 422-423:
> “Additional metrics for lower-case performance can be found in Appendix in Table 11, and for case-sensitive in Tables 9 and 10, respectively.”
> - Line 479: reference to the “Table 13”
>
> For now, we have made our dataset publicly available:
> [https://huggingface.co/datasets/marsianin500/Speech2Latex](https://huggingface.co/datasets/marsianin500/Speech2Latex).
>
> Please feel free to ask additional questions; we will be glad to answer them.

---

### Official Review · Reviewer_Aunz · 2024-11-04

**Soundness:** 2
**Presentation:** 1
**Contribution:** 2
**Rating:** 5
**Confidence:** 4

**Summary:**

This paper introduces a new Speech to Latex dataset. The dataset comprises spoken equations in English and Russian as well as synthetic data. Various systems (ASR + difference LLMs) are evaluated on the dataset and provide initial benchmarks.

**Strengths:**

* the tasks is reasonably well motivated (transcribe some part of a lecture to latex)
* several LLM models are tested

**Weaknesses:**

* The main issue is in the presentation. The paper appears to still be in draft form (missing references, e.g. 214 and 479, Figure 1 is not referenced; results and discussion section seems to be still be in draft form) and does not provide enough details on the methodology.
* The impact of the dataset is unclear for now given that the baseline models that are benchmarked are simply using ASR as a frontend and the main task becomes converting text to latex formula.

**Questions:**

No questions but presentation comments:
214: “in the Appendix in Table ??.” missing reference
416-420: this paragraph seems somewhat out of place and may be a better fit for the introduction
422-423: missing references
479: missing reference

---

> ### Author Response · Authors · 2024-11-22
>
> Thank you for the detailed review. Let us address weak points and answer your questions.
>
> The mentioned uncertainties and miss-printings are fixed:
> - Line 214->242:  reference to the “Table 6”
> - Lines 416-420 moved to the introduction.
> - Lines 422-423:
> “Additional metrics for lower-case performance can be found in Appendix in Table 11, and for case-sensitive in Tables 9 and 10, respectively.”
> - Line 479: reference to the “Table 13”
>
> These errors will be fixed. The clarity and presentation of the revisited text will also be improved.
>
> We disagree that the main task of the work is to convert text to LaTeX; this is merely one approach discussed in the manuscript. In contrast, we explored a multi-modal approach using audio LLMs, where audio is translated directly into LaTeX without explicitly converting it to phonetic text.
>
> For now, we have made our dataset publicly available:
> [https://huggingface.co/datasets/marsianin500/Speech2Latex](https://huggingface.co/datasets/marsianin500/Speech2Latex).
>
> Please feel free to ask additional questions; we will be glad to answer them.

---

> > ### Comment · Reviewer_Aunz · 2024-11-26
> >
> > Thanks for the clarifications on presentation and on the approach and for releasing the dataset. I've raise the score to 5.

---

### Official Review · Reviewer_9HD1 · 2024-11-04

**Soundness:** 2
**Presentation:** 2
**Contribution:** 2
**Rating:** 3
**Confidence:** 5

**Summary:**

The title is a good summary: the authors describe the process of creating an annotated audio-based dataset to help building and evaluating models for automatic conversion of spoken formulas to latex code. The submission adds performance evaluations using different LLM pipelines on this dataset.

**Strengths:**

The idea and motivation is good. Comparing multiple LLM pipelines and NLU-centric metrics is in general good as well.

**Weaknesses:**

The biggest weakness of the submission is that it doesn't help to continue research on this topic as the dataset is not made publicly available. The submission explains some details of the process to create it, but given the complexity of that process and the very brief, vague and unclear description of it between lines 283 and 297, I doubt that any researcher can reproduce these results. The brief description does not even allow to reverse engineer which TTS voices and parameters have been used. In addition, the reader is being left in the dark what XTTSv2 means or why only two/one voices have been used despite possibly more being available?

Another weakness is the description of the used evaluation metrics (lines 320-348). A citation to chrF and chrF++ is missing as well as a definition. Furthermore it is unclear wether chrF or chrF++ has been used. A definition of CER is repeated, but a citation would last there as the definition is simple and well known *plus* it has been widely used over the last 3 decades at least. The unbalanced care in defining the different metrics being used is puzzling.

The analysis of results is limited and in my view not formulated well (lines 392-412). SALMONN-13B consistently shines on all metrics and languages except CER, while Qwen2.5-* has a slight edge over SLAMONN-13B wrt CER.

A few text passages lack clarity:
- line 024: the abstract seems shortened and ends with a performance metric (25%) of unknown unit.
- missing targets of in-text references to sections and tables: line 242, lines 422-423, line 479 (I also check the whole supplementary data)

**Questions:**

I suggest to substantially add clarity, make the dataset publicly available and resubmit as the motivation of the work is good and certainly of interest to a subset of the community.

---

> ### Author Response · Authors · 2024-11-22
>
> Thank you for the detailed review. Let us address weak points and answer your questions.
>
>
> We have chosen not to include links to the data in the submission for anonymity. However, we have made it publicly available while maintaining anonymity thus far [https://huggingface.co/datasets/marsianin500/Speech2Latex](https://huggingface.co/datasets/marsianin500/Speech2Latex).
>
> Our motivation was to verify whether training on artificially generated audio gains similar quality on genuine test speech as training on human-annotated audio. XTTSv2 was chosen for audio generation because this model is publically available with pre-trained weights, generates good-quality audio and allows for voice adjustment (voice conversion). Due to the fact that generating a large number of audio samples using this TTS model is time-consuming, we have chosen to select only a limited number of reference voices for our hypothesis testing purposes.
>
> The references would be expanded in the updated version of the manuscript:
>
> Popović, M. (2015, September). chrF: character n-gram F-score for automatic MT evaluation. In Proceedings of the tenth workshop on statistical machine translation (pp. 392-395).
>
> Popović, M. (2017, September). chrF++: words helping character n-grams. In Proceedings of the second conference on machine translation (pp. 612-618).
>
> We decided to explain CER as a primary metric. Please refer to Appendix A.2 for the details of other metrics. We used both chrF and chrF++ in our experiments.
>
>
> The mentioned uncertainties and miss-printings are fixed:
> - Line 242:  reference to the “Table 6”
> - Lines 422-423:
> “Additional metrics for lower-case performance can be found in Appendix in Table 11, and for case-sensitive in Tables 9 and 10, respectively.”
> - Line 479: reference to the “Table 13”
>
> The undefined metric at the end of an abstract is the character error rate (CER).
>
> The topics regarding metrics definition and the dataset creation procedure will be revised for clarity and simplicity.
>
> Please feel free to ask additional questions; we will be glad to answer them.

---

> ### Comment · Reviewer_9HD1 · 2024-11-23
>
> I admit that based on the naming, I didn't open the file abstract.pdf of the supplementary data, but in fact it contained the (missing) appendices. I recommend to include the appendices in the paper as they do add some clarity. The dataset is made publicly available as described. In order to verify improvements to the clarity of the presentation I recommend to upload a revised version of the submission.
>
> As a consequence I raise my contribution score from 1 to 2 (unfortunately, raising the final score from 3 to 4 isn't possible). Proof of adding clarity to the presentation may help to improve on the presentation score which may raise the final score further to 5.

---

> > ### Comment · Reviewer_9HD1 · 2024-11-26
> > **Please upload a revised version**
> >
> > I reiterate my interest to view a revised version of the submission. According to the CallOfPapers, uploading revisions is explicitly welcomed during the Discussion Phase.

---

### Note · Authors · 2024-11-28

**Comment:**

Dear Reviewers and Committee,

We extend our sincere gratitude for your invaluable assessment of our work.

Taking into careful consideration all of the insightful comments provided, we have reached the decision to withdraw our article temporarily. This choice will enable us to diligently rework and elevate the quality of our paper, aligning it with the standards we aspire to uphold.

Once again, we express our gratitude for your dedicated review and look forward to the opportunity to resubmit an improved version in the future.

Best regards,
Authors

**Withdrawal Confirmation:**

I have read and agree with the venue's withdrawal policy on behalf of myself and my co-authors.